# Fusing Multi-View Scores via Coupling Flows for Time Series Anomaly Detection

## Abstract

Time series anomaly detection is crucial for ensuring system reliability across various applications ranging from industrial monitoring to financial fraud detection. However, two fundamental challenges remain to be addressed: (1) Model bias caused by the inherent diversity of anomaly patterns; (2) Detection inflexibility caused by the scarcity of anomaly labels. We propose MSFlow (Multi-view Score Fusion via Coupling Flows), which constructs a coupling flow-based ensemble capable of modeling complex joint distributions of multi-dimensional scores through invertible transformations. Leveraging this flexible fusion framework, we strategically select four detection perspectives (clustering and reconstruction in both temporal and frequency domains). The coupling flows learn inter-view dependencies while preserving each perspective's unique detection capabilities, achieving effective integration that simple aggregation fails to accomplish. When labels are available, an uncertainty-guided enhancement mechanism identifies high-disagreement regions in ensemble predictions and selectively refines them through a learned soft router, enabling seamless adaptation from unsupervised to semi-supervised operation. Extensive experiments on 10 univariate and 8 multivariate benchmark datasets demonstrate that MSFlow achieves state-of-the-art performance across diverse anomaly types and label availability scenarios.

## 1 Introduction

Time series anomaly detection has become a critical capability across diverse real-world applications, from industrial equipment monitoring and financial fraud detection to fault diagnosis and automotive maintenance systems. The ability to automatically identify anomalous patterns in time series data is essential for ensuring system reliability, operational safety, and service continuity. The increasing complexity and scale of modern time series data demand advanced detection methods that can both capture various anomaly types and adapt to evolving operational conditions. Despite significant advances in anomaly detection techniques, two fundamental challenges continue to limit the performance of existing methods: the inherent diversity of anomaly patterns and the inflexibility in utilizing available supervision signals.

The first challenge stems from the remarkable diversity of time series anomaly types. Temporal anomalies manifest as either point anomalies affecting individual observations (including global and contextual anomalies) or subsequence anomalies spanning continuous segments (seasonal, trend, and shapelet anomalies). Researchers have developed various detection approaches, from statistical methods and classical machine learning to deep learning architectures, demonstrating excellent detection capabilities. However, different anomaly types exhibit vastly different behavioral patterns, and single detection mechanisms—constrained by their theoretical foundations and design biases—struggle

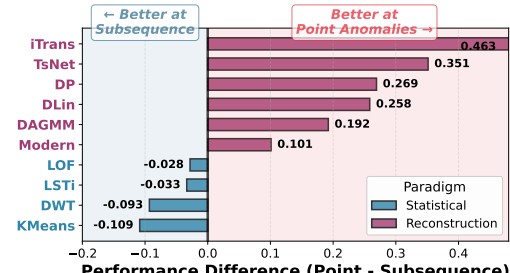

Figure 1: Method-specific detection biases. Positive values indicate better point anomaly detection; negative values indicate subsequence superiority.

to simultaneously capture these diverse characteristics (as shown in the figure B, detailed discussion in the appendix 1). This specialization inevitably limits the applicability of individual methods in complex real-world scenarios. The second challenge arises from the scarcity of anomaly labels in time series scenarios. This label scarcity has popularized unsupervised methods that operate without strict requirements on labeled data, while supervised and semi-supervised methods fail to handle limited annotation scenarios. This creates a fundamental gap: existing methods either completely discard available labels that could enhance detection performance, or require extensive labeling efforts that are impractical in most real-world scenarios. The absence of frameworks that can flexibly adapt to varying label availability prevents practitioners from leveraging valuable supervision signals that naturally accumulate in practical deployments.

We propose MSFlow (*M*ulti-view *S*core Fusion via Coupling *Flow*s), a framework that synergistically addresses the above challenges through a flexible multi-view ensemble architecture and adaptively utilizes available labels. For the first challenge, we develop an ensemble framework based on multiple coupling flows that can model the complex joint distribution of multi-view anomaly scores. Leveraging the flexibility of this ensemble framework, we strategically select four complementary detection perspectives spanning temporal/frequency domains and clustering/reconstruction paradigms, aiming to better cover diverse anomaly types. The coupling flow ensembles learn inter-view dependencies while preserving each view's unique detection capabilities. For the second challenge, we evaluate prediction disagreement among coupling flow ensembles as sample uncertainty—these uncertain samples represent the most valuable learning opportunities. Through uncertainty-guided selective supervision, we employ ranking loss to train a soft routing mechanism that ensures anomaly scores consistently exceed normal scores. This design enables MSFlow to effectively utilize valuable labels when available, while naturally degrading to unsupervised operation when labels are absent and identifying which samples most urgently require annotation.

We demonstrate MSFlow's effectiveness through extensive experiments on 18 diverse benchmarks (10 univariate and 8 multivariate datasets), showing consistent superiority over 20 state-of-the-art baseline methods. MSFlow achieves significant performance improvements when incorporating minimal labeled data and demonstrates effectiveness across multiple anomaly types, validating its practical applicability in various real-world scenarios.

- We propose MSFlow, a multi-view ensemble framework that models the joint distribution of diverse anomaly scores using coupling flows, effectively capturing different types of anomalies through complementary detection perspectives.

- We introduce uncertainty-guided selective supervision that leverages prediction disagreement to identify valuable annotation samples, enabling flexible adaptation from unsupervised to semi-supervised settings without requiring predefined label ratios.

- Extensive experiments on 18 benchmarks demonstrate MSFlow's consistent superiority over 20 state-of-the-art methods, particularly achieving significant performance improvements with minimal labeled data.

## 2 RELATED WORK

### 2.1 ANOMALY DETECTION PARADIGMS

**Statistical Distribution-based Methods**  Statistical methods detect anomalies by finding data that differs from normal patterns (Chandola et al., 2009). Classic approaches include distance-based methods (Ahmed et al., 2020; Schubert et al., 2017; Li et al., 2003; Yi & Yoon, 2020; Ruff et al., 2018; Breunig et al., 2000; Guo et al., 2003; Yeh et al., 2016; Liu et al., 2008; Hariri et al., 2019) that identify outliers based on their proximity to normal data groups, density estimation (Goldstein & Dengel, 2012; Inoue & Shintani, 2006; Zong et al., 2018) that spots samples in low-density regions, and clustering-based methods (Yairi et al., 2001; He et al., 2003) that group normal patterns and identify deviations. Tests from the TAB benchmark (Qiu et al., 2025) show these traditional methods often beat deep learning at finding pattern anomalies. This is likely because they directly measure geometry and are sensitive to local structures. Recent work uses diffusion models (Zhang et al., 2025; Wyatt et al., 2022; Gathiaka et al., 2016) to learn complex patterns. These models spot anomalies by checking how well they can reconstruct data.

**Reconstruction-based Methods**   Reconstruction methods work on a simple idea: models trained on normal data make larger errors when they see abnormal patterns (Sakurada & Yairi, 2014; Kingma & Welling, 2013). These methods have evolved with deep learning. They started with basic autoencoders (Zhou et al., 2021; Niu et al., 2020; Gong et al., 2019; Sakurada & Yairi, 2014). Then came recurrent networks (Niu et al., 2020; Su et al., 2019; Bhatnagar et al., 2021). Now we have attention-based models (Attention; Xu et al., 2021). Deep learning methods, especially Transformers (Xu et al., 2021; Yang et al., 2023; Wu et al., 2022; Nie et al., 2023; Liu et al., 2024a), have succeeded greatly due to their powerful representation advantages and are good at finding point anomalies. They learn complex features that catch single-point errors and local context problems (Qiu et al., 2025). Large pre-trained models (Chang et al., 2025; Goswami et al., 2024; Gao et al., 2024; Liu et al., 2024b) make this even better through transfer learning. However, these complex models can fail with simple outliers (Zeng et al., 2023). They also struggle with pattern anomalies because they focus too much on local quality and miss long-term shifts.

**Discriminative Learning Methods**   Discriminative methods learn boundaries between normal and abnormal regions directly. They do not just rely on statistics or reconstruction quality. One-class classification (Ruff et al., 2018; Carmona et al., 2021; Schölkopf et al., 1999) builds tight boundaries around normal data in feature spaces. Contrastive learning (Yang et al., 2023; Yue et al., 2022; Zhuang et al., 2025) creates useful features by checking consistency across different views. It doesn't need anomaly labels. But these methods have problems. Without real anomalies for training, they must create fake ones that might not match real patterns. The learned boundaries might only work for specific anomaly types seen during training. New anomaly patterns are hard to handle.

### 2.2 Joint Temporal-Frequency Domain Analysis

Frequency analysis finds anomalies that time-based methods miss. These include broken cycles, warped harmonics, and spectral problems (Ren et al., 2019; Thill et al., 2017; Hyndman & Athanasopoulos, 2018). Different anomalies show up better in different domains. Seasonal anomalies appear clearly in frequency domain as broken patterns. Trend anomalies show better in time domain as long-term shifts (Qiu et al., 2025). Early work focused on single-variable spectral analysis (Feng et al., 2021; Ren et al., 2019). It used frequency maps and partial Fourier transforms to find anomalies. Modern methods (Wu et al., 2022; Yang et al., 2023; Zhang et al., 2022; 2019; Nam et al., 2024a; Wu et al., 2024) try to combine time and frequency analysis. They face several challenges: time and frequency details do not always match (Nam et al., 2024b), high-frequency information gets lost during processing, and cross-channel relationships are hard to model (Wu et al., 2024). Current fusion methods just combine features simply or merge them late. They fail to model the complex links between time and frequency scores.

## 3 Methodology

Consider a time series $\mathcal{X} = \langle x_1, x_2, \ldots, x_T \rangle$, where each point $x_t \in \mathbb{R}^d$ represents a $d$-dimensional measurement at time $t$. The training data $\mathcal{X}$ primarily contains normal behavior patterns (Chandola et al., 2009; Lai et al., 2021). Our goal is to build a detector that identifies anomalies in test sequences $\mathcal{X}_{\text{test}}$, producing binary labels $\mathcal{Y}_{\text{test}} = \langle y_1, y_2, \ldots, y_{T'} \rangle$ where $y_t \in \{0, 1\}$ indicates normal (0) or abnormal (1) behavior.

### 3.1 Framework Overview

MSFlow employs a staged approach to time series anomaly detection, centered on probabilistic modeling of multi-view anomaly scores through coupling flows. The framework consists of three main components: (1) an unsupervised foundation that models the joint distribution of anomaly scores from multiple detection perspectives using an ensemble of coupling flows; (2) a supervised enhancement mechanism that leverages prediction disagreement to identify and improve uncertain regions when labeled data is available; (3) an anomaly scoring system that adaptively combines unsupervised and supervised signals based on detection confidence. We provide detailed algorithmic descriptions of the three core components in Algorithms 1–3 in the Appendix for readers interested in implementation details.

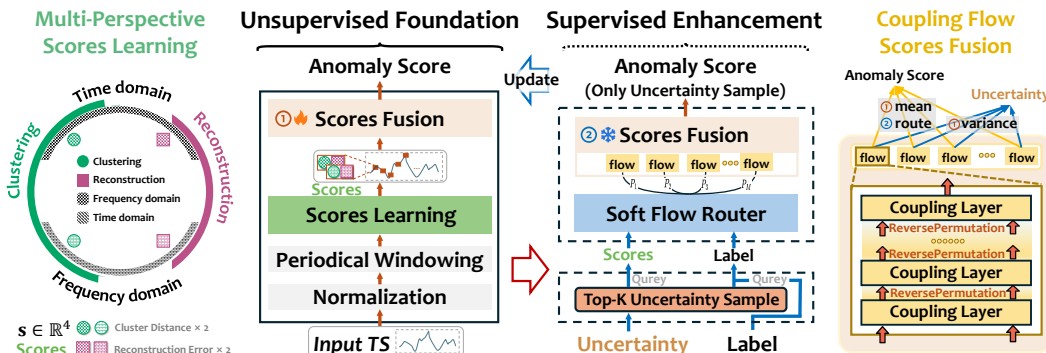

Figure 2: Architecture of MSFlow: A Multi-View Time Series Anomaly Detection Framework. The framework processes time series through adaptive windowing, parallel multi-view scoring across temporal and frequency domains, coupling flow-based probabilistic fusion, and optional uncertainty-guided enhancement.

Figure 2 illustrates the overall architecture. The framework first extracts anomaly scores from four complementary perspectives—clustering and reconstruction methods in both temporal and frequency domains—to capture diverse anomaly characteristics. These multi-view scores are then processed by an ensemble of coupling flows that learn their joint distribution through invertible transformations, producing fused anomaly scores. The key advantage of using multiple flows is that their prediction disagreement (variance) naturally quantifies detection uncertainty. When labeled data becomes available, this uncertainty guides selective supervision: during training, it identifies the Top-K most uncertain samples for training the soft router with ranking loss. During inference, the learned router is applied to all test samples when available, dynamically adjusting fusion weights based on the input score patterns. This design ensures the framework operates effectively in purely unsupervised settings while seamlessly incorporating supervision when available to enhance detection performance.

## 3.2 Unsupervised Foundation

**Multi-View Score Learning**  The framework begins with standardization to ensure comparability across dimensions, followed by adaptive windowing that segments data into fixed-length windows $\mathbf{W}_i \in \mathbb{R}^{W \times d}$. The window size $W$ can be automatically determined through FFT-based periodicity detection to align with natural data cycles, or set manually based on domain knowledge.

For each window, we compute anomaly scores from four complementary perspectives (see Appendix C for detailed rationale behind this design choice):

- **Temporal domain clustering:** We apply K-means clustering to identify $K$ typical temporal patterns in the training data (Yairi et al., 2001; He et al., 2003). For each test window, we compute its Euclidean distance to the nearest cluster center. Windows that fall far from all learned clusters likely represent anomalous patterns not seen during training.

- **Temporal domain reconstruction:** We employ a Transformer model with inverted tokenization (Liu et al., 2024a; Nie et al., 2023), where each variable's time series serves as a token rather than each time point. The model learns to reconstruct normal temporal patterns through self-attention mechanisms (Xu et al., 2021). High reconstruction errors indicate deviations from learned normal dependencies.

- **Frequency domain clustering:** After applying FFT to convert windows to frequency domain, we perform K-means clustering on the magnitude spectra. This captures typical frequency patterns and identifies anomalies with unusual spectral characteristics that temporal analysis might miss (Ren et al., 2019).

- **Frequency domain reconstruction:** A specialized Transformer operates directly on frequency representations (both magnitude and phase), learning to reconstruct normal spectral patterns. This method excels at detecting subtle frequency anomalies like broken period-

icities, harmonic distortions, or abnormal frequency components (Thill et al., 2017; Nam et al., 2024a; Wu et al., 2024).

This multi-view approach leverages the observation that different anomaly types manifest distinctively across domains and detection paradigms. Point anomalies typically show high reconstruction errors, while pattern anomalies are better captured by clustering distances (Liu et al., 2008). Since we use overlapping windows with stride $s$, each timestamp may appear in multiple windows. We aggregate these window-level scores to point-level through averaging: $s_t^{(v)} = \frac{1}{|\mathcal{W}_t|} \sum_{w \in \mathcal{W}_t} s_w^{(v)}$, where $\mathcal{W}_t$ denotes all windows containing timestamp $t$ (Yeh et al., 2016; Tatbul et al., 2018). The resulting score matrix $\mathbf{S}_t \in \mathbb{R}^{d \times 4}$ for each timestamp provides a comprehensive characterization of potential anomalies from multiple perspectives.

**Coupling Flow-Based Score Fusion**   Traditional score aggregation methods (such as averaging or weighted combination) fail to capture the complex dependencies between different detection methods (Pevnỳ, 2016; Zong et al., 2018). Since the entire process operates without anomaly labels, we address this limitation through coupling flows, which perform self-supervised modeling of the complete joint distribution of multi-dimensional scores via invertible transformations (see Appendix C for motivation).

Before fusion, all scores are standardized within each variable to zero mean and unit variance, ensuring comparability across different detection methods (Bhatnagar et al., 2021). Each coupling flow $f_e : \mathbb{R}^{d \times 4} \to \mathbb{R}^{d \times 4}$ consists of $L_c$ coupling layers that progressively transform the complex score distribution to a standard Gaussian distribution in latent space. We naturally organize the score matrix into temporal $\mathbf{T} \in \mathbb{R}^{d \times 2}$ (clustering and reconstruction scores) and frequency $\mathbf{F} \in \mathbb{R}^{d \times 2}$ (clustering and reconstruction scores) partitions. Each coupling layer applies conditional affine transformations through an alternating scheme:

$$\mathbf{T}^{(\ell+1)} = \mathbf{T}^{(\ell)}, \quad \mathbf{F}^{(\ell+1)} = \mathbf{F}^{(\ell)} \odot \exp(\mathbf{s}_\theta(\mathbf{T}^{(\ell)})) + \mathbf{t}_\theta(\mathbf{T}^{(\ell)}) \tag{1}$$

$$\mathbf{F}^{(\ell+1)} = \mathbf{F}^{(\ell)}, \quad \mathbf{T}^{(\ell+1)} = \mathbf{T}^{(\ell)} \odot \exp(\mathbf{s}_\phi(\mathbf{F}^{(\ell)})) + \mathbf{t}_\phi(\mathbf{F}^{(\ell)}) \tag{2}$$

where $\mathbf{s}$ and $\mathbf{t}$ are scale and translation parameters generated by coupling networks $\theta$ and $\phi$. This alternating structure enables bidirectional modeling of dependencies between temporal and frequency domains, allowing the model to learn complex relationships such as correlations between specific frequency anomalies and temporal patterns.

**Ensemble Strategy**   We train an ensemble of $E$ coupling flows with different random initializations and data subsets to create diversity. Each flow independently optimizes the log-likelihood of observed score patterns:

$$\mathcal{L}_{\text{flow}}^{(e)} = \frac{1}{B} \sum_{i=1}^{B} \left[ \frac{1}{2} \|\mathbf{f}_e(\mathbf{S}_i)\|_F^2 - \log | \det J_{f_e}(\mathbf{S}_i)| + C \right] \tag{3}$$

where $J_{f_e}$ denotes the Jacobian determinant ensuring invertibility, and $C$ is a normalization constant.

The ensemble serves three critical purposes: (1) *Robustness*—model averaging reduces individual flow biases and improves generalization; (2) *Uncertainty quantification*—prediction disagreement naturally identifies samples where the unsupervised detection is least confident, providing valuable guidance for selective supervision; (3) *Selective refinement*—enabling targeted improvement of uncertain regions without affecting confident predictions. The uncertainty for each sample is quantified as:

$$u_t = \text{Var}_{e \in \{1, ..., E\}}[r_e(\mathbf{S}_t)] \tag{4}$$

where $r_e(\mathbf{S}_t) \in [0, 1]$ is the percentile rank of the sample's likelihood relative to the training distribution, providing a normalized uncertainty measure robust to likelihood scale variations.

## 3.3 SUPERVISED ENHANCEMENT

When labeled data becomes available, MSFlow refines its detection capabilities through uncertainty-guided enhancement without requiring complete model retraining (Schmidl et al., 2022; Jacob et al., 2020). The key insight is that regions of high ensemble disagreement represent the most valuable improvement opportunities where supervision can have maximum impact.

**Uncertainty-Guided Sample Selection** We leverage the ensemble's prediction variance $u_t$ to identify samples where supervision would be most beneficial. By selecting the Top-K samples with highest uncertainty (top $p_{\text{train}}\%$), we ensure that limited labels are strategically used to address the model's blind spots rather than redundantly confirming already-confident predictions. This active learning strategy maximizes the information gained from scarce labeled data.

**Soft Flow Router** For the selected high-uncertainty training samples, we introduce a lightweight routing network $g_\psi : \mathbb{R}^{d \times 4} \to \mathbb{R}^E$ that takes the original 4-dimensional anomaly scores $\mathbf{S}_t$ as input—specifically the clustering and reconstruction scores from both temporal and frequency domains—and learns to dynamically assign weights to each ensemble member. The router is trained exclusively on high-uncertainty samples using a ranking loss that encourages correct relative ordering between anomalous and normal samples:

$$\mathcal{L}_{\text{rank}} = \frac{1}{|\mathcal{P}|} \sum_{(i,j) \in \mathcal{P}} \log\left(1 + \exp\left(\phi(\mathbf{S}_j) - \phi(\mathbf{S}_i)\right)\right) \tag{5}$$

where $\mathcal{P}$ contains all valid pairs with anomaly $i$ ($y_i = 1$) and normal sample $j$ ($y_j = 0$) from the high-uncertainty subset. The routed score $\phi(\mathbf{S}) = \sum_{e=1}^E w_e(\mathbf{S}) \cdot r_e(\mathbf{S})$ combines individual flow predictions using learned weights $w_e$ computed via temperature-controlled softmax. Once trained, the router is applied to all test samples, allowing it to adaptively select the most reliable flows for different score patterns across the entire test set.

## 3.4 ANOMALY SCORING

The framework converts coupling flow likelihoods to interpretable anomaly scores through percentile ranking. During training, we store the likelihood distribution of normal samples as a reference. At test time, each flow computes the negative log-likelihood for a test sample and converts it to a percentile rank $r_e(\mathbf{S}_t)$ by determining what percentage of training samples have lower likelihoods. This normalization ensures scores from different flows are comparable and provides an intuitive interpretation: a rank of 0.95 means the sample is more anomalous than 95% of training data.

The final anomaly score depends on whether supervised enhancement is available:

$$\text{AnomalyScore}(\mathbf{S}_t) = \begin{cases} \phi(\mathbf{S}_t) & \text{if supervised enhancement is trained} \\ \frac{1}{E} \sum_{e=1}^E r_e(\mathbf{S}_t) & \text{otherwise (pure unsupervised)} \end{cases} \tag{6}$$

where $\phi(\mathbf{S}_t) = \sum_{e=1}^E w_e(\mathbf{S}_t) \cdot r_e(\mathbf{S}_t)$ represents the router-enhanced scores using learned weights. When labeled data is available for training the soft router, the enhanced scoring is applied to all test samples, leveraging the learned weighting mechanism to improve detection performance. In the absence of labeled data, the framework defaults to simple ensemble averaging, maintaining strong unsupervised capabilities.

## 4 EXPERIMENT

### 4.1 EXPERIMENTAL SETTINGS

#### 4.1.1 DATASETS

We conduct comprehensive evaluation on 18 diverse time series anomaly detection benchmarks covering both 10 univariate datasets (GAIA, GHL, KDD21, MGAB, NASA-MSL, NAB, OPPORTU-

NITY, NASA-SMAP, SVDB, YAHOO) and 8 multivariate datasets (LTDB, MITDB, MSL, SMD, NYC, PSM, SMAP, SVDB), more details of the benchmark datasets are included in Appendix Table 5 in Appendix A.1.

### 4.1.2 BASELINES

We compare against 18 state-of-the-art methods spanning three major anomaly detection paradigms: statistical distribution-based approaches (6 methods), reconstruction-based methods (10 methods), and discriminative learning approaches (2 methods), as detailed in Table 1. The baselines encompass non-learning statistical methods, traditional machine learning algorithms, and modern deep learning architectures.

Table 1: Baseline methods for comparison

| Paradigm | Method | Abbrev. | Key Technology |
|---|---|---|---|
| Statistical Distribution-based | ● LOF (Breunig et al., 2000) | LOF | Density |
| | ● KNN (Ramaswamy et al., 2000) | KNN | K-nearest Neighbors |
| | ● Isolation Forest (Liu et al., 2008) | IF | Isolation Tree |
| | ● HBOS (Goldstein & Dengel, 2012) | HBOS | Histogram |
| | ● OC-SVM (Schölkopf et al., 1999) | OCSVM | One-class Classification |
| | ● K-Means (Yairi et al., 2001) | KMeans | K-Means Clustering |
| Reconstruction-based | ● VAE (Kingma & Welling, 2013) | VAE | Variational Autoencoder |
| | ● DAGMM (Zong et al., 2018) | DAGMM | Deep Autoencoder + GMM |
| | ● DeepPoint (Bhatnagar et al., 2021) | DP | Multilayer Perceptron |
| | ● TranAD (Tuli et al., 2022) | TranAD | Transformer + Adversarial |
| | ● Anomaly Transformer (Xu et al., 2021) | ATrans | Anomaly Attention |
| | ● DualTF (Nam et al., 2024a) | DualTF | Time-Frequency Fusion |
| | ● PatchTST (Nie et al., 2023) | Patch | Channel Independent + Patch |
| | ● ModernTCN (Luo & Wang, 2024) | Modern | Enhanced TCN |
| | ● DLinear (Zeng et al., 2023) | DLin | Linear + Decomposition |
| | ● iTransformer (Liu et al., 2024a) | iTrans | Inverted Transformer |
| Discriminative Learning | ● DCdetector (Yang et al., 2023) | DC | Contrastive Learning |
| | ● ContraAD (Zhuang et al., 2025) | ConAD | Contrastive Learning |

● Non Learning, ● Machine Learning, ● Deep Learning

### 4.1.3 SETUP

We utilize TAB (Qiu et al., 2025) code for unified evaluation, with all baseline results also derived from TAB. To ensure fair comparison and demonstrate the robustness of our approach, we employ a single set of unified hyperparameters for all univariate datasets and another single set for all multivariate datasets, without dataset-specific tuning. We adopt Label-based metric: *Affiliated F1-score* (Huet et al., 2022) and Score-based metric: Area under the Receiver Operating Characteristics Curve (ROC) *AUROC* (Fawcett, 2006) as evaluation metrics. More implementation details are presented in the Appendix A.3.

### 4.2 MAIN RESULTS

Tablese 2 present the evaluation results on univariate and multivariate datasets, respectively. MS-Flow achieves the best performance on the majority of benchmarks, demonstrating strong generalization across diverse anomaly detection scenarios. For univariate time series, MSFlow obtains particularly impressive results on challenging datasets with complex temporal patterns, while maintaining consistent superiority in ranking quality. On multivariate benchmarks, our method delivers robust performance by effectively capturing cross-dimensional dependencies through the coupling flow mechanism. Compared to recent deep learning methods like ModernTCN, DualTF, and Anomaly Transformer, MSFlow demonstrates more stable performance across different data characteristics—while these baseline methods often excel on specific datasets but struggle on others, our approach maintains competitive results throughout.

**Impact of Supervised Enhancement.** To evaluate the effectiveness of uncertainty-guided label utilization, we analyze MSFlow's performance with varying amounts of labeled data. Table 4 demonstrates the impact of different $p_{train}$ percentages on detection performance.

Table 2: Average A-R (AUC-ROC) and Aff-F (Affiliated-F1) accuracy measures for all datasets. The best results are highlighted in **Red**, and the second-best results are Blue.

| Dataset | Metric | MSFlow | Modern | DualTF | iTrans | ATrans | DC | TranAD | ConAD | Patch | DLin | VAE | DAGMM | KNN | KMeans | OCSVM | IF | DP | LOF | HBOS |
|---|---|---|---|---|---|---|---|---|---|---|---|---|---|---|---|---|---|---|---|---|
| **Univariate Datasets** | | | | | | | | | | | | | | | | | | | | |
| GAIA | Aff-F | 0.871 | 0.904 | 0.753 | **0.907** | 0.697 | 0.711 | 0.790 | 0.683 | 0.902 | 0.897 | 0.727 | 0.598 | 0.759 | 0.781 | 0.725 | 0.731 | 0.673 | 0.737 | 0.689 |
| | A-R | **0.872** | 0.821 | 0.564 | 0.823 | 0.446 | 0.411 | 0.756 | 0.486 | 0.819 | 0.801 | 0.748 | 0.648 | 0.821 | 0.738 | 0.726 | 0.695 | 0.755 | 0.817 | 0.687 |
| GHL | Aff-F | **0.975** | 0.723 | 0.682 | 0.709 | 0.652 | 0.669 | 0.577 | 0.667 | 0.735 | 0.730 | 0.555 | 0.465 | 0.679 | 0.598 | 0.667 | 0.687 | 0.667 | 0.679 | 0.638 |
| | A-R | **0.998** | 0.493 | 0.374 | 0.498 | 0.476 | 0.498 | 0.020 | 0.500 | 0.500 | 0.528 | 0.087 | 0.020 | 0.751 | 0.747 | 0.751 | 0.500 | 0.888 | 0.751 | 0.662 |
| KDD21 | Aff-F | **0.895** | 0.718 | 0.698 | 0.767 | 0.698 | 0.686 | 0.723 | 0.685 | 0.713 | 0.726 | 0.652 | 0.640 | 0.678 | 0.822 | 0.669 | 0.675 | 0.667 | 0.675 | 0.666 |
| | A-R | **0.887** | 0.534 | 0.569 | 0.605 | 0.499 | 0.493 | 0.528 | 0.502 | 0.529 | 0.538 | 0.540 | 0.538 | 0.638 | 0.778 | 0.590 | 0.554 | 0.532 | 0.584 | 0.564 |
| MGAB | Aff-F | **0.793** | 0.678 | 0.675 | 0.691 | 0.671 | 0.673 | 0.670 | 0.667 | 0.669 | 0.668 | 0.664 | 0.620 | 0.666 | 0.688 | 0.666 | 0.664 | 0.667 | 0.667 | 0.665 |
| | A-R | **0.773** | 0.561 | 0.576 | 0.501 | 0.500 | 0.504 | 0.592 | 0.522 | 0.559 | 0.567 | 0.555 | 0.545 | 0.560 | 0.605 | 0.550 | 0.487 | 0.515 | 0.509 | 0.586 |
| MSL | Aff-F | **0.922** | 0.896 | 0.766 | 0.865 | 0.743 | 0.757 | 0.902 | 0.673 | 0.883 | 0.903 | 0.686 | 0.609 | 0.775 | 0.894 | 0.795 | 0.658 | 0.673 | 0.772 | 0.733 |
| | A-R | **0.803** | 0.726 | 0.537 | 0.724 | 0.469 | 0.506 | 0.546 | 0.500 | 0.727 | 0.710 | 0.652 | 0.609 | 0.673 | 0.725 | 0.666 | 0.561 | 0.636 | 0.614 | 0.607 |
| NAB | Aff-F | 0.828 | 0.838 | 0.782 | 0.832 | 0.751 | 0.714 | **0.858** | 0.701 | 0.848 | 0.850 | 0.739 | 0.620 | 0.730 | 0.818 | 0.722 | 0.711 | 0.680 | 0.695 | 0.744 |
| | A-R | **0.640** | 0.612 | 0.550 | 0.614 | 0.469 | 0.506 | 0.546 | 0.506 | 0.605 | 0.608 | 0.556 | 0.572 | 0.563 | 0.626 | 0.584 | 0.535 | 0.531 | 0.545 | 0.570 |
| OPP | Aff-F | **0.788** | 0.733 | 0.710 | 0.709 | 0.728 | 0.679 | 0.762 | 0.770 | 0.732 | 0.737 | 0.674 | 0.628 | 0.656 | 0.734 | 0.654 | 0.654 | 0.667 | 0.671 | 0.653 |
| | A-R | 0.664 | 0.215 | 0.411 | 0.224 | **0.665** | 0.503 | 0.559 | 0.504 | 0.217 | 0.304 | 0.514 | 0.578 | 0.461 | 0.285 | 0.487 | 0.242 | 0.260 | 0.545 | 0.522 |
| SMAP | Aff-F | **0.967** | 0.913 | 0.686 | 0.928 | 0.764 | 0.760 | 0.851 | 0.680 | 0.906 | 0.906 | 0.595 | 0.569 | 0.717 | 0.886 | 0.698 | 0.655 | 0.674 | 0.711 | 0.647 |
| | A-R | **0.899** | 0.694 | 0.535 | 0.752 | 0.480 | 0.487 | 0.520 | 0.502 | 0.707 | 0.615 | 0.596 | 0.586 | 0.594 | 0.802 | 0.586 | 0.513 | 0.655 | 0.551 | 0.526 |
| SVDB | Aff-F | **0.833** | 0.733 | 0.745 | 0.718 | 0.685 | 0.685 | 0.730 | 0.682 | 0.727 | 0.728 | 0.683 | 0.611 | 0.710 | 0.735 | 0.698 | 0.672 | 0.682 | 0.709 | 0.700 |
| | A-R | **0.780** | 0.573 | 0.586 | 0.576 | 0.504 | 0.460 | 0.549 | 0.500 | 0.569 | 0.580 | 0.570 | 0.549 | 0.520 | 0.683 | 0.567 | 0.544 | 0.531 | 0.516 | 0.553 |
| YAHOO | Aff-F | **0.914** | 0.793 | 0.748 | 0.897 | 0.671 | 0.629 | 0.755 | 0.678 | 0.800 | 0.825 | 0.615 | 0.453 | 0.660 | 0.675 | 0.897 | 0.669 | 0.664 | 0.653 | 0.653 |
| | A-R | **0.936** | 0.804 | 0.467 | 0.891 | 0.477 | 0.490 | 0.697 | 0.488 | 0.828 | 0.843 | 0.756 | 0.707 | 0.734 | 0.589 | 0.639 | 0.924 | 0.862 | 0.754 | 0.678 |
| **Multivariate Datasets** | | | | | | | | | | | | | | | | | | | | |
| LTDB | Aff-F | **0.794** | 0.777 | 0.455 | 0.756 | 0.702 | 0.710 | 0.790 | 0.752 | 0.767 | 0.767 | 0.670 | 0.653 | 0.768 | 0.724 | 0.754 | 0.644 | 0.724 | 0.763 | 0.759 |
| | A-R | **0.833** | 0.619 | 0.589 | 0.579 | 0.505 | 0.494 | 0.590 | 0.576 | 0.587 | 0.599 | 0.610 | 0.545 | 0.632 | 0.738 | 0.615 | 0.559 | 0.551 | 0.621 | 0.611 |
| MITDB | Aff-F | **0.925** | 0.803 | 0.843 | 0.806 | 0.700 | 0.689 | 0.848 | 0.694 | 0.813 | 0.722 | 0.713 | 0.689 | 0.709 | 0.834 | 0.700 | 0.709 | 0.678 | 0.700 | 0.687 |
| | A-R | **0.767** | 0.663 | 0.693 | 0.679 | 0.424 | 0.503 | 0.691 | 0.488 | 0.676 | 0.583 | 0.667 | 0.682 | 0.692 | 0.692 | 0.689 | 0.618 | 0.598 | 0.627 | 0.681 |
| MSL | Aff-F | **0.738** | 0.726 | 0.258 | 0.705 | 0.685 | 0.674 | 0.724 | 0.671 | 0.714 | 0.723 | 0.642 | 0.723 | 0.696 | 0.580 | 0.641 | 0.584 | 0.677 | 0.701 | 0.680 |
| | A-R | **0.645** | 0.621 | 0.499 | 0.589 | 0.502 | 0.500 | 0.478 | 0.548 | 0.621 | 0.601 | 0.530 | 0.569 | 0.623 | 0.603 | 0.524 | 0.524 | 0.488 | 0.557 | 0.574 |
| SMD | Aff-F | 0.794 | **0.839** | 0.679 | 0.817 | c | 0.674 | 0.789 | 0.023 | 0.830 | 0.834 | 0.450 | 0.023 | 0.696 | 0.686 | 0.742 | 0.626 | 0.674 | 0.682 | 0.629 |
| | A-R | **0.749** | 0.721 | 0.703 | 0.745 | 0.509 | 0.500 | 0.663 | 0.491 | 0.736 | 0.708 | 0.641 | 0.527 | 0.716 | 0.713 | 0.602 | 0.664 | 0.672 | 0.645 | 0.626 |
| NYC | Aff-F | **0.946** | 0.693 | 0.676 | 0.683 | 0.732 | 0.674 | 0.796 | 0.714 | 0.805 | 0.725 | 0.767 | 0.694 | 0.644 | 0.646 | 0.667 | 0.648 | 0.668 | 0.652 | 0.675 |
| | A-R | **0.808** | 0.696 | 0.723 | 0.594 | 0.859 | 0.526 | 0.676 | 0.443 | 0.667 | 0.699 | 0.661 | 0.573 | 0.466 | 0.833 | 0.456 | 0.475 | 0.476 | 0.464 | 0.446 |
| PSM | Aff-F | 0.739 | 0.823 | 0.725 | **0.855** | 0.634 | 0.671 | 0.748 | 0.648 | 0.836 | 0.832 | 0.295 | 0.463 | 0.695 | 0.735 | 0.531 | 0.620 | 0.694 | 0.694 | 0.658 |
| | A-R | 0.635 | 0.587 | 0.544 | 0.583 | 0.499 | 0.499 | 0.635 | 0.533 | 0.583 | 0.559 | 0.642 | 0.637 | **0.744** | 0.732 | 0.619 | 0.542 | 0.539 | 0.730 | 0.714 |
| SMAP | Aff-F | 0.551 | 0.616 | 0.674 | 0.577 | **0.692** | 0.681 | 0.538 | 0.430 | 0.629 | 0.607 | 0.487 | 0.430 | 0.630 | 0.517 | 0.503 | 0.512 | 0.681 | 0.642 | 0.509 |
| | A-R | 0.468 | 0.434 | 0.465 | 0.409 | 0.501 | 0.500 | 0.369 | 0.364 | 0.441 | 0.391 | 0.412 | 0.573 | **0.629** | 0.405 | 0.393 | 0.487 | 0.429 | 0.626 | 0.585 |
| SVDB | Aff-F | **0.896** | 0.746 | 0.585 | 0.746 | 0.701 | 0.711 | 0.752 | 0.694 | 0.787 | 0.727 | 0.697 | 0.692 | 0.733 | 0.820 | 0.723 | 0.708 | 0.692 | 0.734 | 0.735 |
| | A-R | **0.913** | 0.593 | 0.555 | 0.636 | 0.491 | 0.464 | 0.577 | 0.533 | 0.651 | 0.610 | 0.570 | 0.564 | 0.595 | 0.839 | 0.585 | 0.533 | 0.527 | 0.580 | 0.568 |

The results reveal a characteristic learning curve with diminishing returns: adding just 5% of labeled data yields substantial improvements (average gain of 0.033), while increasing from 50% to 100% provides minimal additional benefit (gain of only 0.004). This pattern validates our uncertainty-guided sample selection strategy—by focusing on the most informative samples identified through ensemble disagreement, MSFlow achieves near-optimal performance with minimal supervision. The rapid initial improvement demonstrates that the soft router effectively learns to correct the ensemble's systematic errors in high-uncertainty regions, while the plateau at higher label percentages suggests that confident predictions require no correction. This label-efficient behavior is particularly valuable in practical deployments where obtaining comprehensive labels is costly or infeasible.

## 4.3 MODEL ANALYSIS

We conduct comprehensive experiments to analyze MSFlow from multiple perspectives. This section presents ablation studies to validate each component's contribution and visualizations of anomaly detection capabilities across different anomaly types. Additional analyses are provided in the appendix, including detailed performance breakdown for five fundamental anomaly types (Appendix B) and extensive parameter sensitivity analysis.

Table 3: Performance (AUC-ROC) with varying amounts of labeled data for supervised enhancement. The baseline (0%) represents pure unsupervised MSFlow.

| Dataset | Train AR (%) | MSFlow (Unsupervised) | $p_{test}$ (%) | | | | |
|---|---|---|---|---|---|---|---|
| | | | 1 | 5 | 20 | 50 | 100 |
| NAB | 6.55 | 0.640 | 0.680 | 0.700 | 0.710 | 0.700 | 0.700 |
| OPPORTUNITY | 19.11 | 0.664 | 0.664 | 0.664 | 0.664 | 0.664 | 0.664 |
| SVDB | 5.48 | 0.780 | 0.810 | 0.820 | 0.820 | 0.820 | 0.820 |
| YAHOO | 5.48 | 0.936 | 0.941 | 0.945 | 0.944 | 0.945 | 0.945 |
| LTDB | 11.72 | 0.833 | 0.847 | 0.853 | 0.858 | 0.860 | 0.857 |
| SVDB* | 5.48 | 0.913 | 0.914 | 0.915 | 0.914 | 0.915 | 0.915 |

**Impact of Supervised Enhancement.** To evaluate the effectiveness of uncertainty-guided label utilization, we analyze MSFlow's performance with varying amounts of labeled data. Table 4 demonstrates the impact of different $p_{train}$ percentages on detection performance.

The results reveal a characteristic learning curve with diminishing returns: incorporating a small fraction of labeled data yields substantial performance improvements, while further increasing the label percentage provides progressively smaller gains. This pattern validates our uncertainty-guided sample selection strategy—by focusing on the most informative samples identified through ensem-

ble disagreement, MSFlow achieves near-optimal performance with minimal supervision. The rapid initial improvement demonstrates that the soft router effectively learns to correct the ensemble's systematic errors in high-uncertainty regions, while the plateau at higher label percentages suggests that confident predictions require no correction. This label-efficient behavior is particularly valuable in practical deployments where obtaining comprehensive labels is costly or infeasible.

**Anomaly Type Visualization.** Figure 3 demonstrates MSFlow's multi-view scoring mechanism across five fundamental anomaly types: Global Point and Contextual Point (from YAHOO), and Seasonal, Trend, and Shapelet (from SVDB). The visualization reveals complementary detection

Table 4: Performance (AUC-ROC) with varying amounts of labeled data for supervised enhancement. The baseline (0%) represents pure unsupervised MSFlow.

| Dataset | Train AR (%) | MSFlow (Unsupervised) | $p_{test}$ (%) | | | | |
|---|---|---|---|---|---|---|---|
| | | | 1 | 5 | 20 | 50 | 100 |
| NAB | 6.55 | 0.6402 | 0.6807 | 0.7003 | 0.7106 | 0.7004 | 0.7002 |
| OPPORTUNITY | 19.11 | 0.6643 | 0.6641 | 0.6645 | 0.6642 | 0.6644 | 0.6643 |
| SVDB | 5.48 | 0.7801 | 0.8103 | 0.8206 | 0.8204 | 0.8205 | 0.8203 |
| YAHOO | 5.48 | 0.9362 | 0.9413 | 0.9452 | 0.9447 | 0.9451 | 0.9453 |
| LTDB | 11.72 | 0.8334 | 0.8471 | 0.8532 | 0.8584 | 0.8603 | 0.8572 |
| SVDB* | 5.48 | 0.9131 | 0.9142 | 0.9154 | 0.9146 | 0.9152 | 0.9151 |

patterns across different views. For point anomalies (first two columns), temporal reconstruction scores produce sharp, localized peaks precisely at anomaly positions, while frequency scores remain relatively stable. Conversely, for subsequence anomalies (last three columns), frequency reconstruction scores maintain sustained elevation throughout the entire anomalous intervals, effectively capturing extended pattern violations that temporal methods might fragment. Notably, the seasonal anomaly shows frequency clustering scores dominating the detection, while trend and shapelet anomalies benefit from both temporal and frequency reconstruction. This view-specific sensitivity validates our multi-perspective design—each detection method contributes uniquely based on anomaly characteristics, enabling MSFlow to achieve robust detection across diverse anomaly types through coupling flow-based fusion.

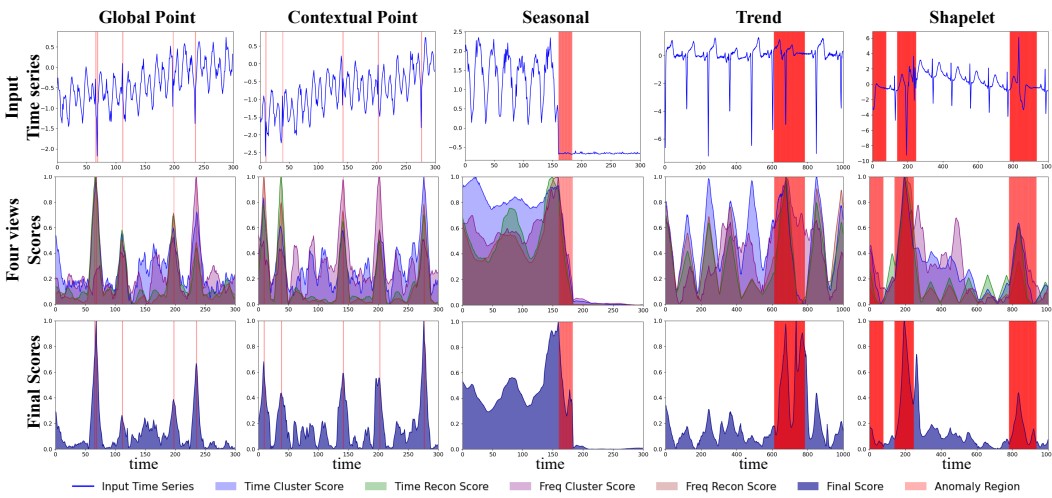

Figure 3: Multi-view anomaly detection across five types. Rows show: input time series, four view scores (temporal clustering, temporal reconstruction, frequency clustering, frequency reconstruction), and final fused scores. Red regions indicate ground-truth anomalies.

## 5 CONCLUSION

In this paper, we presented MSFlow, a multi-view anomaly detection framework that addresses two fundamental challenges in time series analysis: the diversity of anomaly patterns and the flexible utilization of available supervision. Through coupling flow-based score fusion, MSFlow effectively integrates complementary detection perspectives from temporal and frequency domains, capturing both point and subsequence anomalies that single methods often miss. The uncertainty-guided enhancement mechanism enables the framework to adaptively leverage labeled data when available while maintaining strong unsupervised performance. Extensive experiments on 18 diverse benchmarks demonstrate MSFlow achieves state-of-the-art performance.

ETHICS STATEMENT

Our work exclusively uses publicly available benchmark datasets that contain no personally identifiable information. The proposed anomaly detection framework is designed for beneficial applications in system reliability and safety monitoring. No human subjects were involved in this research.

REPRODUCIBILITY STATEMENT

To ensure reproducibility, we provide: (1) Complete implementation details including all hyperparameters in Appendix A.3; (2) Source code and scripts are provided in an anonymous repository at [https://anonymous.4open.science/r/MSflow-020D]; (3) All experiments use the standard public benchmark (TAB) protocols and implementations with documented preprocessing steps; (4) Fixed random seeds for all stochastic components.

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

# A  EXPERIMENTAL DETAILS

## A.1  DATASET

Table 5: Summary of evaluation datasets

| Type | Dataset | Domain | Dim | Series | Avg Total Length | Avg Test Length | Avg AR (%) | Train AR (%) |
|---|---|---|---|---|---|---|---|---|
| Univariate | GAIA () | AIOps | 1 | 184 | 9,777 | 8,799 | 1.26 | 0.19 |
| | GHL (Filonov et al., 2016) | Machinery | 1 | 1 | 200,001 | 180,001 | 0.43 | 0.00 |
| | KDD21 (Keogh, 2021) | Multiple | 1 | 243 | 65,903 | 47,294 | 0.58 | 0.00 |
| | MGAB (Thill et al., 2020) | Mackey-Glass | 1 | 6 | 100,000 | 90,000 | 0.20 | 0.00 |
| | NASA-MSL (Benecki et al., 2021) | Spacecraft | 1 | 22 | 5,167 | 2,897 | 3.85 | 0.00 |
| | NAB (Ahmad et al., 2017) | Multiple | 1 | 45 | 7,115 | 3,557 | 9.84 | 6.55 |
| | OPPORTUNITY (Roggen et al., 2010) | Movement | 1 | 462 | 31,359 | 28,223 | 4.12 | 19.11 |
| | NASA-SMAP (Benecki et al., 2021) | Spacecraft | 1 | 35 | 10,539 | 7,930 | 2.20 | 0.00 |
| | SVDB (Greenwald, 1990) | Health | 1 | 52 | 230,400 | 207,360 | 4.87 | 5.48 |
| | YAHOO (Laptev et al., 2015) | Multiple | 1 | 346 | 1,570 | 785 | 0.63 | 0.46 |
| Multivariate | LTDB (A. L. Goldberger & Stanley, 2000) | Health | 2 | 5 | 100,000 | 87,285 | 15.57 | 11.72 |
| | MITDB (A. L. Goldberger & Stanley, 2000) | Health | 2 | 6 | 141,667 | 106,250 | 2.72 | 0.00 |
| | MSL (Hundman et al., 2018) | Spacecraft | 55 | 1 | 132,046 | 73,729 | 5.88 | 0.00 |
| | SMD (Su et al., 2019) | Server Machine | 38 | 1 | 1,416,825 | 708,420 | 2.08 | 0.00 |
| | NYC (Cui et al., 2016) | Transport | 3 | 1 | 17,520 | 4,416 | 0.57 | 0.00 |
| | PSM (Abdulaal et al., 2021) | Server Machine | 25 | 1 | 220,322 | 87,841 | 11.07 | 0.00 |
| | SMAP (Hundman et al., 2018) | Spacecraft | 25 | 1 | 562,800 | 427,617 | 9.72 | 0.00 |
| | SVDB (Greenwald, 1990) | Health | 2 | 6 | 110,133 | 86,373 | 3.14 | 5.48 |

AR: anomaly ratio

In order to comprehensively evaluate the performance of MSflow, we evaluate 18 diverse time series anomaly detection benchmarks covering both 10 univariate datasets and 8 multivariate datasets, spanning multiple domains including spacecraft telemetry, server monitoring, healthcare, movement analysis, and industrial systems. These datasets exhibit varied characteristics in terms of dimensionality (1-55 channels), sequence lengths (785-1,416,825 points), test anomaly ratios (0.20%-15.57%), and training anomaly ratios (0.00%-19.11%), ensuring robust validation across different operational contexts. Table 5 lists statistics of the 18 datasets. As shown in Table 5, we conduct comprehensive evaluation on 18 diverse time series anomaly detection benchmarks covering both univariate (10 datasets) and multivariate (8 datasets) scenarios

As shown in Table 5, we conduct comprehensive evaluation on 18 diverse time series anomaly detection benchmarks covering both univariate (10 datasets) and multivariate (8 datasets) scenarios

## A.2  METRICS

The metrics we support can be divided into two categories: Score-based and Label-based. Label-based metrics includes Accuracy ($Acc$), Precision ($P$), Recall ($R$), F1-score ($F1$), Range-Precision ($R$-$P$), Range-Recall ($R$-$R$), Range-F1-score ($R$-$F$) (Tatbul et al., 2018), Precision@k, Affiliated-Precision ($Aff$-$P$), Affiliated-Recall ($Aff$-$R$), and Affiliated-F1-score ($Aff$-$F$) (Huet et al., 2022). Score-based metrics includes the Area Under the Precision-Recall Curve ($A$-$P$) (Davis & Goadrich, 2006), the Area under the Receiver Operating Characteristics Curve ($A$-$R$) (Fawcett, 2006), the Range Area Under the Precision-Recall Curve ($R$-$A$-$P$), the Range Area under the Receiver Operating Characteristics Curve ($R$-$A$-$R$) (Paparrizos et al., 2022), the Volume Under the Surface of Precision-Recall ($V$-$PR$), and the Volume Under the Surface of Receiver Operating Characteristic ($V$-$ROC$) (Paparrizos et al., 2022). While we report Affiliated-F1 and AUC-ROC in the main paper for clarity, complete results across all metrics for every dataset and baseline method are provided in our repository at [`https://anonymous.4open.science/r/MSflow-020D`]. More implementation details are presented in the Appendix A.3.

## A.3  IMPLEMENTATION DETAILS

**Dataset Splitting.** We follow the original train-test splits when provided. Otherwise, we use a 50%-50% split for training and testing. For baseline methods requiring validation, we extract the last 20% from the training set as validation data. Our method does not use validation sets and utilizes the full training portion.

**Training Configuration.** All baselines use their official implementations with default hyperparameters. We set batch size to 128 (or 64 when encountering OOM issues) and apply early stopping with patience of 10 epochs for deep learning methods. For memory-intensive models, we adopt stride-doubling to maintain computational feasibility.

**Threshold Selection.** Due to threshold sensitivity in anomaly detection, we evaluate across $\tau \in \{0.1, 0.5, 1, 2, 3, 5, 10, 15, 20, 25\}$ (percentiles) for both univariate and multivariate settings, reporting the best performance for each method-dataset pair. All metrics are reported as percentages.

**Computational Setup.** Experiments are conducted using Python 3.8 with PyTorch 1.12.0 (Paszke et al., 2019) on NVIDIA Tesla-A800 GPUs. Random seeds are fixed at 42 for reproducibility. We employ point-adjust (PA) evaluation (Xu et al., 2018) and use bias parameter 0.2 for segment-based metrics following (Huet et al., 2022).

## B  METHOD-SPECIFIC DETECTION BIAS ANALYSIS

To empirically validate the existence of method-specific biases, we conducted an extensive analysis across 35 anomaly detection methods spanning three paradigms, evaluating them on univariate datasets categorized by their primary anomaly patterns: point anomalies (including global and contextual anomalies), subsequence anomalies (seasonal, trend, and shapelet anomalies), and mixed cases combining multiple types. Table 6 lists all evaluated methods with the 18 baseline methods from our main experiments highlighted in blue. All experiments followed fair and consistent evaluation protocols from the TAB benchmark (Qiu et al., 2025), enabling systematic analysis of detection capabilities across different anomaly granularities.

Table 6: Evaluated methods across three detection paradigms

| Paradigm | Methods (Abbreviation) |
|---|---|
| Statistical Distribution-based (14 methods) | LOF (Breunig et al., 2000), KNN (Ramaswamy et al., 2000), CBLOF (He et al., 2003), KMeans (Yairi et al., 2001), LSTi (Yeh et al., 2016), ZMS (Bhatnagar et al., 2021), HBOS (Goldstein & Dengel, 2012), Stat (Bhatnagar et al., 2021), DWT (Thill et al., 2017), OCSVM (Schölkopf et al., 1999), LODA (Pevnỳ, 2016), IF (Liu et al., 2008), EIF (Hariri et al., 2019), COF (Tang et al., 2002) |
| Reconstruction-based (19 methods) | ARIMA (Hyndman & Athanasopoulos, 2018), SARIMA (Greis et al., 2018), Torsk (Heim & Avery, 2019), LSTM (Bhatnagar et al., 2021), DP (Bhatnagar et al., 2021), SR (Ren et al., 2019), PCA (Shyu et al., 2003), DAGMM (Zong et al., 2018), iTrans (Liu et al., 2024a), TsNet (Wu et al., 2022), ATrans (Xu et al., 2021), Patch (Nie et al., 2023), Modern (Luo & Wang, 2024), TranAD (Tuli et al., 2022), DualTF (Nam et al., 2024a), AE (Sakurada & Yairi, 2014), VAE (Kingma & Welling, 2013), NLin (Zeng et al., 2023), DLin (Zeng et al., 2023) |
| Discriminative Learning (2 methods) | DC (Yang et al., 2023), ConAD (Zhuang et al., 2025) |

Note: Blue indicates baseline methods from main experiments.

**Key Findings.** The combined analysis of Figures 4 and 5 reveals three important observations about method-specific detection biases:

**(1) High-performance methods exhibit specialized detection capabilities.** Figure 4 shows that top-performing methods display pronounced specialization patterns. For instance, iTrans achieves exceptional performance on global anomalies with strong positive bias, yet shows reduced effectiveness and negative bias in seasonal detection. Similarly, DP demonstrates strong capabilities for contextual anomalies while showing limitations in subsequence detection, and KMeans excels at seasonal anomalies but shows negative bias for global detection. The heat map indicates that methods with higher average performance generally display more pronounced specialization patterns, whereas lower-ranked methods tend toward more uniform but modest performance across categories.

**(2) Paradigm characteristics influence detection preferences.** The performance distributions in Figure 5 indicate systematic tendencies across paradigms. Deep learning methods (iTrans, TsNet, DLin) frequently appear among top performers for global anoma-

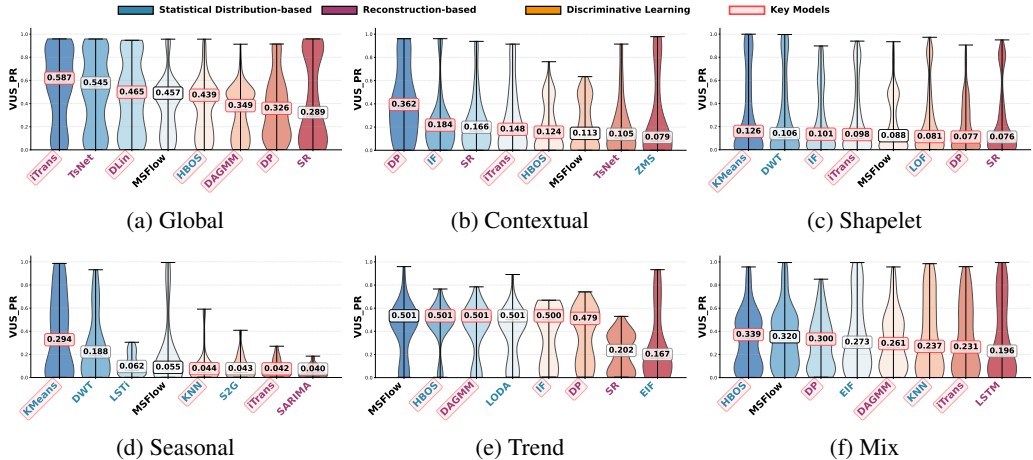

(a) Global      (b) Contextual      (c) Shapelet

(d) Seasonal      (e) Trend      (f) Mix

Figure 5: Performance distributions across anomaly types. Top performers shown per category, illustrating method specialization patterns.

lies but are less prominent in subsequence categories. Statistical approaches demonstrate complementary strengths—KMeans and DWT dominate seasonal and trend detection, while HBOS excels at shapelet patterns. The heat map further supports this observation: reconstruction methods tend toward positive biases for point anomalies and negative biases for subsequence types, while statistical methods often exhibit the opposite pattern.

**(3) Mixed anomalies present universal challenges.** Figure 5f reveals that even the best-performing methods achieve only modest performance on mixed anomalies, with multiple methods (MSFlow, HBOS, DAGMM, LODA) plateauing at similar levels. This performance ceiling, considerably lower than peak performances on single-type categories, suggests that scenarios combining multiple anomaly patterns pose significant challenges that current single-method approaches struggle to address effectively.

**MSFlow's Superior Performance.** These observations validate MSFlow's multi-view fusion strategy. By leveraging coupling flows to combine complementary detection perspectives, MSFlow successfully addresses individual method limitations. Empirical results confirm MSFlow achieves the highest average performance and consistently ranks among the top performers across all six anomaly categories. Unlike specialized methods that excel in specific domains but struggle elsewhere (as shown by the extreme biases in Figure 4), MSFlow maintains balanced performance with minimal bias variations across categories, demonstrating that principled fusion of complementary perspectives provides a more robust solution than pursuing universal capability within single architectures.

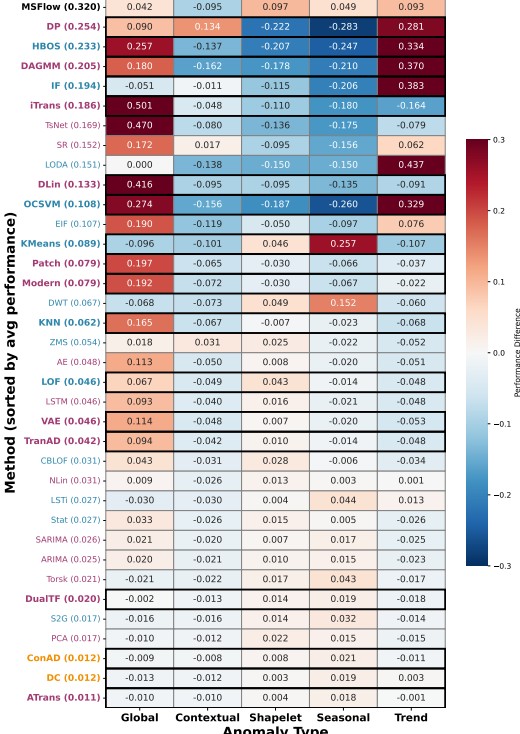

Figure 4: Performance bias matrix. Methods sorted by average performance. Each cell shows relative performance difference: specific type versus mean of others. Warm colors indicate positive bias (better on this type), cool colors indicate negative bias.

## C  RATIONALE
## FOR MULTI-VIEW SELECTION
## AND COUPLING FLOW FUSION

The empirical findings from our method-specific bias analysis (Section B) provide strong motivation for our architectural choices in MSFlow. The analysis revealed that reconstruction methods excel at detecting point anomalies but struggle with subsequence patterns, while clustering approaches demonstrate the opposite strength—effectively capturing pattern-level deviations but missing subtle point variations. Furthermore, the complementary nature of temporal and frequency domains is evident: seasonal and periodic anomalies manifest clearly in frequency space (as shown by KMeans and DWT's dominance in seasonal detection), while trend and contextual anomalies are better captured in the time domain (where iTrans and TsNet excel). This naturally leads to our selection of four detection views: temporal clustering captures subsequence deviations in time series patterns, temporal reconstruction identifies point-wise violations of learned sequential dependencies, frequency clustering detects anomalous spectral patterns and periodicity breaks, and frequency reconstruction captures subtle harmonic distortions. The coupling flow fusion mechanism is essential because the relationships between these views are complex and non-linear—for instance, a seasonal anomaly might simultaneously manifest as a clustering outlier in frequency domain and a reconstruction error in temporal domain, but with varying intensities depending on the anomaly's characteristics. Simple averaging or weighted combination would fail to capture these intricate dependencies, while coupling flows, through their invertible transformations and alternating conditioning between temporal and frequency partitions, can learn the joint distribution of multi-view scores and model how different views complement or contradict each other for various anomaly types. This design directly addresses the key finding that even top-performing individual methods plateau on mixed anomalies, as the coupling flow can leverage the collective strength of all views to handle complex scenarios where multiple anomaly types coexist.

When labeled data becomes available, Algorithm 3 selectively enhances predictions through uncertainty-guided refinement. The key insight is focusing limited labeled data on regions where the unsupervised ensemble shows high disagreement, preserving confident predictions while improving ambiguous cases through learned routing.

## D  PARAMETER SENSITIVITY.

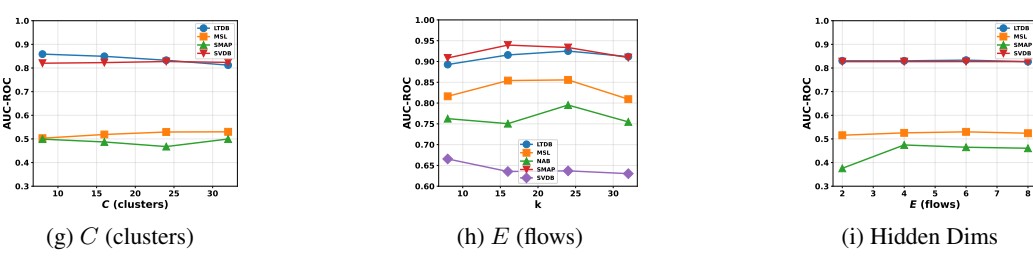

(g) $C$ (clusters)  (h) $E$ (flows)  (i) Hidden Dims

Figure 6: Parameter sensitivity analysis results. The AUC-ROC performance is evaluated across different hyperparameter settings for four benchmark datasets (LTDB, MSL, SMAP, SVDB).

We investigate the sensitivity of MSFlow to key hyperparameters that directly impact the multi-view scoring and fusion mechanisms. Figure 6 presents the performance variation across different parameter configurations on four benchmark datasets. The parameter sensitivity analysis reveals that MSFlow exhibits graceful performance curves across its hyperparameter space, indicating a well-designed architecture that doesn't rely on precise tuning to achieve strong results.

## E  ALGORITHM DESCRIPTION

The MSFlow framework operates through three core algorithmic components that process data sequentially. We present the key algorithms following their execution order.

**Algorithm 1** Multi-View Score Learning

1: **Input:** Time series $\mathcal{X} \in \mathbb{R}^{T \times d}$, window size $W$
2: **Output:** Multi-view score matrix $\mathbf{S} \in \mathbb{R}^{T \times d \times 4}$
3:
4: **// Extract windows with detected or default period**
5: $W \leftarrow \text{DetectPeriod}(\mathcal{X})$ if adaptive else $W_{\text{default}}$
6: $\mathcal{W} \leftarrow \text{SlidingWindows}(\mathcal{X}, W, \text{stride})$
7:
8: **for** window $\mathbf{W}_i \in \mathcal{W}$ **do**
9:     **for** dimension $v = 1$ to $d$ **do**
10:         **// Temporal domain**
11:         $s_1 \leftarrow \text{ClusteringDistance}(\mathbf{W}_i[:, v])$
12:         $s_2 \leftarrow \text{TransformerReconError}(\mathbf{W}_i[:, v])$
13:
14:         **// Frequency domain**
15:         $\mathbf{F} \leftarrow |\text{FFT}(\mathbf{W}_i[:, v])|$
16:         $s_3 \leftarrow \text{ClusteringDistance}(\mathbf{F})$
17:         $s_4 \leftarrow \text{ReconstructionError}(\mathbf{F})$
18:
19:         Store $[s_1, s_2, s_3, s_4]$ for window $i$, dimension $v$
20:     **end for**
21: **end for**
22:
23: **// Aggregate and normalize**
24: $\mathbf{S} \leftarrow \text{AverageOverlappingWindows(scores)}$
25: $\mathbf{S} \leftarrow \text{Normalize}(\mathbf{S})$ {Zero mean, unit variance}
26: **Return $\mathbf{S}$** =0

Algorithm 1 extracts multi-view scores that serve as input to our fusion framework. By computing complementary scores across temporal and frequency domains using both clustering and reconstruction paradigms, we capture diverse anomaly characteristics that individual methods might miss.

**Algorithm 2** Coupling Flow-Based Score Fusion

1: **Input:** Multi-view score matrix $\mathbf{S} \in \mathbb{R}^{N \times d \times 4}$ for $N$ samples
2: **Output:** Ensemble anomaly scores $\mathcal{A} \in \mathbb{R}^N$
3:
4: **// Train ensemble of coupling flows**
5: **for** $e = 1$ to $E$ **do**
6:     Initialize flow $f_e$ with $L_c$ coupling layers
7:     **for** training iteration **do**
8:         $\mathbf{T}, \mathbf{F} \leftarrow \text{Partition}(\mathbf{S})$ {Temporal and frequency domains}
9:
10:         **// Alternating coupling transformations**
11:         **for** layer $\ell = 1$ to $L_c$ **do**
12:             **if** $\ell$ is even **then**
13:                 $\mathbf{s}, \mathbf{t} \leftarrow \text{CouplingNet}_\theta(\mathbf{T})$ {Compute transform params}
14:                 $\mathbf{F} \leftarrow \mathbf{F} \odot \exp(\mathbf{s}) + \mathbf{t}$ {Transform frequency using temporal}
15:             **else**
16:                 $\mathbf{s}, \mathbf{t} \leftarrow \text{CouplingNet}_\theta(\mathbf{F})$
17:                 $\mathbf{T} \leftarrow \mathbf{T} \odot \exp(\mathbf{s}) + \mathbf{t}$ {Transform temporal using frequency}
18:             **end if**
19:         **end for**
20:
21:         **// Optimize likelihood**
22:         $\mathcal{L} \leftarrow \frac{1}{2}\|f_e(\mathbf{S})\|^2 - \log|\det J_{f_e}(\mathbf{S})|$
23:         Update parameters $\theta_e$ via gradient descent
24:     **end for**
25: **end for**
26:
27: **// Compute ensemble anomaly scores**
28: **for** sample $i = 1$ to $N$ **do**
29:     $\mathcal{A}[i] \leftarrow \frac{1}{E} \sum_{e=1}^{E} \text{PercentileRank}(f_e(\mathbf{S}_i))$
30: **end for**
31: **Return $\mathcal{A}$** =0

The coupling flow mechanism (Algorithm 2) represents our core innovation for fusing multi-view scores. By alternately conditioning transformations between temporal and frequency domains, the flow captures complex interdependencies that simple aggregation methods miss. The ensemble approach with $E$ independent flows provides robustness and uncertainty estimates.

---

**Algorithm 3** Supervised Enhancement (Optional)

---

1: **Input:** Ensemble predictions $\{r_e(\mathbf{S})\}_{e=1}^{E}$, labels $\mathcal{Y}$
2: **Output:** Enhanced scores with learned router
3:
4: **// Identify uncertain regions for training**
5: **for** each sample $i$ **do**
6:     $u_i \leftarrow \mathrm{Var}_{e \in \{1,\ldots,E\}}[r_e(\mathbf{S}_i)]$ {Ensemble disagreement}
7: **end for**
8:
9: **// Train router on high-uncertainty samples**
10: $\mathcal{H} \leftarrow \{i : u_i > \mathrm{Percentile}(u, p_{\mathrm{train}})\}$ {High uncertainty set}
11: Initialize router $g_\psi : \mathbb{R}^{d \times 4} \to \mathbb{R}^E$
12: **for** training iteration **do**
13:     Sample pairs $(i, j)$ where $i \in \mathcal{H} \cap \{y = 1\}, j \in \mathcal{H} \cap \{y = 0\}$
14:     $\mathbf{w}_i \leftarrow \mathrm{Softmax}(g_\psi(\mathbf{S}_i))$ {Adaptive flow weights}
15:     $\phi_i \leftarrow \sum_e w_i[e] \cdot r_e(\mathbf{S}_i)$ {Weighted ensemble}
16:     $\mathcal{L}_{\mathrm{rank}} \leftarrow \log(1 + \exp(\phi_j - \phi_i))$ {Ranking loss}
17:     Update $\psi$ via gradient descent
18: **end for**
19:
20: **// Apply to all samples during inference**
21: **Return** $\phi(\mathbf{S}) = \sum_e g_\psi(\mathbf{S})[e] \cdot r_e(\mathbf{S})$ for all test samples =0

---

When labeled data becomes available, Algorithm 3 selectively enhances predictions through uncertainty-guided refinement. The key insight is focusing limited labeled data on regions where the unsupervised ensemble shows high disagreement, preserving confident predictions while improving ambiguous cases through learned routing.

# F    USE OF AI ASSISTANTS

AI tools were used for grammar checking and formatting only. All research content is original.

