# OpenReview forum: "MSFlow: Fusing Multi-Perspective Scores through Coupling Flows for Time Series Anomaly Detection"
_ICLR.cc/2026/Conference — ICLR 2026 Conference Withdrawn Submission_

### Official Review · Reviewer_dJd5 · 2025-10-16

**Soundness:** 1
**Presentation:** 3
**Contribution:** 3
**Rating:** 4
**Confidence:** 4

**Summary:**

This paper proposes a novel time-series anomaly detection framework that aggregates anomaly scores from multiple perspectives, including clustering-based and reconstruction-based views across both temporal and frequency domains. The approach also incorporates an uncertainty-guided enhancement mechanism, which leverages limited labeled samples in a semi-supervised fashion to strengthen detection capability. The proposed model is evaluated extensively on 18 benchmark datasets, covering both univariate and multivariate time series, and demonstrates promising overall performance.

**Strengths:**

The paper is generally well written and clearly structured, with a coherent methodology and logical flow. The idea of aggregating anomaly scores from different perspectives to form a multi-view ensemble framework is interesting and has clear practical value. The incorporation of selective supervision through an uncertainty-guided mechanism is also an interesting direction. The experimental section appears comprehensive, including a variety of baselines and datasets.

**Weaknesses:**

The main weakness of the paper is the lack of a fair comparison with existing semi-supervised anomaly detection methods. Since one of the core components of the proposed model is explicitly semi-supervised (the uncertainty-guided enhancement), it is essential to compare against established semi-supervised methods such as Deep-SAD [1] and its variants. Without these comparisons, it is difficult to judge whether the improvement comes from the proposed design or simply from leveraging supervision unavailable to unsupervised baselines.

Additionally, the paper does not include comparisons with methods that also perform aggregation of multiple anomaly scores, such as W2AD [2] or related approaches.

Finally, the paper could benefit from a deeper discussion on the feasibility and practicality of its selective sample-labeling mechanism in real-world anomaly detection. The motivation for how samples are selected and labeled is not entirely clear, especially considering that anomaly detection typically involves very limited and noisy labeling. Clarifying how this labeling process fits operationally into the anomaly detection pipeline would help make the contribution more convincing.

[1] Ruff, Lukas, et al. "Deep semi-supervised anomaly detection." arXiv preprint arXiv:1906.02694 (2019).

[2] Alnegheimish, Sarah, et al. "M $^ 2$ AD: Multi-Sensor Multi-System Anomaly Detection through Global Scoring and Calibrated Thresholding." arXiv preprint arXiv:2504.15225 (2025).

**Questions:**

Have the authors tried benchmarking against semi-supervised models such as Deep-SAD?

How does the aggregation weighting adapt when the component detectors (e.g., clustering and reconstruction) disagree?

---

### Official Review · Reviewer_sBYW · 2025-11-01

**Soundness:** 2
**Presentation:** 2
**Contribution:** 2
**Rating:** 2
**Confidence:** 2

**Summary:**

The paper proposes an anomaly detection scheme that leverages coupling flows to model the joint distribution of multiple anomaly scores, resulting in improved anomaly detection.  The paper also proposes a scheme to leverage labelled data where available.

**Strengths:**

The proposed scheme is evaluated on 18 benchmarks and compared against 18 baselines, and the results appear promising.

The scheme fuses four raw anomaly scores---windowed temporal, frequency-based, clustering, and prediction---using an invertible coupling flow. The results suggest it outperforms prior schemes that simply average scores to make the final detection.

**Weaknesses:**

The exposition is clear; however, the paper should articulate its novelty relative to prior work.  Is this the first work that uses an ensemble technique for anomaly detection?

I didn’t find Figure 2 informative---its contribution to understanding is unclear. Consider replacing it with a pipeline schematic that maps each equation to its stage (inputs -> dynamics -> scoring -> decision).

Please include an ablation on how the ‘top 4’ raw anomaly signals were chosen and how sensitive performance is to this choice.  Add leave-one-out and add-one-in ablations over the 4 signals to quantify each signal’s marginal contribution.

Test robustness to normalization, window size, and dynamics parameters; include detection delay and false-alarm rate.

The paper would benefit from more specific information about how raw scores were computed.

**Questions:**

Can you please highlight the novel aspect of this work in light of prior approaches that fuse anomaly scores or uses ensemble methods for anomaly detection?  Or is this the first such scheme?

What sort of computational resources are needed to 1) train this scheme and 2) deploy it in practice?

---

### Official Review · Reviewer_FmnY · 2025-11-06

**Soundness:** 3
**Presentation:** 3
**Contribution:** 2
**Rating:** 6
**Confidence:** 3

**Summary:**

The paper proposes MSFlow, a framework that extracts four complementary perspectives (temporal/frequency × clustering/reconstruction), fuses their scores with coupling flows to model the joint score distribution, and when labels exist, applies an uncertainty-guided soft router trained with a ranking loss to selectively refine predictions. The method targets both the diversity of anomaly types and limited labels, and reports SOTA results on a set of benchmarks.

**Strengths:**

1. Clear problem framing and motivation. The paper articulates biases of single-paradigm detectors and motivates multi-view fusion well.

2. Label-efficient enhancement. The uncertainty-guided sample selection and routing are simple and sensible; the method degrades to unsupervised when labels are absent.

3. Thorough evaluation. Experiments span both univariate and multivariate datasets using TAB; baselines cover statistical, reconstruction and discriminative families.

4. Ablation on supervision budget. Paper shows robust gains with small label budgets; qualitative visualizations illustrate view complementarity across anomaly types.

5. Reproducibility. The paper provides implementation details and an anonymous repo; fixed seeds are noted.

**Weaknesses:**

1. Novelty. The core ingredients—multi-view scoring, flow-based density modeling, and pairwise ranking—are known. The contribution is the assembly for TSAD plus a soft router. The paper would benefit from a deeper theoretical or empirical analysis of why coupling flows are superior to lighter-weight fusers.

2. Comparison to other types of multi-view score fusion is missing. What is a real impact of coupling flows?

3. Complexity & latency. Training multiple view-specific models plus an ensemble of coupling flows and a router may be heavy. There’s little discussion of compute, memory, and inference latency.

4. Generalization across anomaly regimes. Results are overall strong, but the paper’s narrative would be tighter with a per-type breakdown (point vs. contextual vs. subsequence) showing where MSFlow helps most and where it fails

5. Statistical testing. I didn’t see significance tests across datasets; given many baselines, paired tests and critical-difference diagrams would strengthen claims.

6. Operational details. How are window sizes and FFT settings chosen per dataset? The text mentions adaptive windowing, but more concrete guidance and sensitivity plots would help practitioners.

**Questions:**

1. Why coupling flows over alternatives? Please compare to other types of fusion if possible.

2. Complexity: Provide FLOPs/latency and memory vs baselines.

3. Robustness: How does MSFlow behave under covariate shift (seasonality drift), missing values, or heavy noise?

---

### Note · Authors · 2026-01-15

I have read and agree with the venue's withdrawal policy on behalf of myself and my co-authors.